# The Role of lncRNAs in Gene Expression Regulation through mRNA Stabilization

**DOI:** 10.3390/ncrna7010003

**Published:** 2021-01-05

**Authors:** Maialen Sebastian-delaCruz, Itziar Gonzalez-Moro, Ane Olazagoitia-Garmendia, Ainara Castellanos-Rubio, Izortze Santin

**Affiliations:** 1Department of Genetics, Physical Anthropology and Animal Physiology, University of the Basque Country, 48940 Leioa, Spain; maialen.sebastian@ehu.eus (M.S.-d.); ane.olazagoitia@ehu.eus (A.O.-G.); ainara.castellanos@ehu.eus (A.C.-R.); 2Biocruces Bizkaia Health Research Institute, 48903 Barakaldo, Spain; itziar.gonzalezm@ehu.eus; 3Department of Biochemistry and Molecular Biology, University of the Basque Country, 48940 Leioa, Spain; 4Ikerbasque, Basque Foundation for Science, 48009 Bilbao, Spain; 5CIBER de Diabetes y Enfermedades Metabólicas Asociadas (CIBERDEM), Instituto de Salud Carlos III, 28029 Madrid, Spain

**Keywords:** long non-coding RNA, mRNA stability, RNA binding protein, microRNA, gene expression

## Abstract

mRNA stability influences gene expression and translation in almost all living organisms, and the levels of mRNA molecules in the cell are determined by a balance between production and decay. Maintaining an accurate balance is crucial for the correct function of a wide variety of biological processes and to maintain an appropriate cellular homeostasis. Long non-coding RNAs (lncRNAs) have been shown to participate in the regulation of gene expression through different molecular mechanisms, including mRNA stabilization. In this review we provide an overview on the molecular mechanisms by which lncRNAs modulate mRNA stability and decay. We focus on how lncRNAs interact with RNA binding proteins and microRNAs to avoid mRNA degradation, and also on how lncRNAs modulate epitranscriptomic marks that directly impact on mRNA stability.

## 1. Introduction

Gene expression and translation is influenced by messenger RNA (mRNA) stability in almost all living organisms. mRNA from bacterial cells can last from seconds to more than one hour, but on average it stays functional between 1 and 3 min [1,2]. Conversely, the lifetime of mammalian mRNA ranges from a couple of minutes to even days, making eukaryotic mRNA more stable than bacterial mRNA. However, from bacteria to mammals, mRNA lifetime needs to be finely regulated in order to enable correct cell homeostasis [1]. The control of the abundance of a particular mRNA fluctuates to adapt to environmental changes, cell growth, differentiation, or to adjust to an unfamiliar situation [3,4]. In this line, the regulation of mRNA stability is essential for tissues and organs exposed to stress signals, such as starvation, infection, inflammation, toxins, or tissue invasion by immune cells [5,6].

The levels of mRNA molecules in the cell are determined by a balance between production and decay [7,8]. Maintaining an accurate balance is crucial for the correct function of a wide variety of biological processes and for the maintenance of an appropriate cellular homeostasis. Many variables such as primary and secondary structure, translation rate and location, among others, influence mRNA stability [5,9,10], and thus minor changes in the structure or the sequence of mRNA molecules might directly influence their half-life.

Eukaryotic mRNAs are transcribed in the nucleus, they are capped (7-methylguanosine cap in 5′end), spliced, polyadenylated (poly(A) tail in 3′end), and lastly, mature mRNAs are exported to the cytoplasm where they are translated into the corresponding polypeptides [1]. Once in the cytoplasm, the 5′cap and the 3′tail serve to attract specific protein complexes that regulate mRNA stability, via protecting mRNA molecules from the attack of ribonucleases and decapping enzymes [1,9,11].

In the last few years, significant progress has been made towards the understanding of mRNA degradation and stability. In general, the decay of mRNA molecules in eukaryotic cells starts with the deadenylation and/or decapping of the mature mRNA, followed by degradation carried out by exonucleases [12,13,14,15]. However, the regulation of mRNA stability depends largely on how a three step process (deadenylation, decapping, and degradation) is modulated by regulatory factors, and thus these factors should be taken into account when analyzing the regulation of mRNA stability. Indeed, several studies have pointed out the key role of RNA-binding proteins and miRNAs in the regulation of this process [7,9,16,17]. In addition, long non-coding RNAs (lncRNAs) are emerging as prominent regulators of mRNA stability and decay [3,4,18,19,20,21]. LncRNAs are RNA molecules without protein coding potential with lengths exceeding 200 nucleotides [22]. They play important roles in biological processes such as chromatin remodeling, transcriptional activation and interference, RNA processing, and mRNA translation [23]. Regarding their mechanisms of action, different models have been proposed, including functioning as signal, decoy, scaffold, guide, and enhancer RNAs [24]. Importantly, the expression of lncRNAs occurs in a cell-, tissue-, and species-specific manner, and accumulating evidence suggests that different splice variants of individual lncRNAs are also expressed in a cell-, tissue-, and species-specific way [25].

In this review we provide an overview of the main molecular mechanisms by which lncRNAs modulate mRNA stability and gene expression (Figure 1). A detailed description of how lncRNAs interact with target mRNAs, RNA binding proteins or miRNAs to avoid mRNA degradation is provided and a brief explanation on how lncRNAs modulate epitranscriptomic changes to impact on mRNA stability is also described.

## 2. LncRNAs Affecting mRNA Stability via miRNA Blockage

Recent several studies have been focused on the analysis of the cross-talk between non-coding and coding RNAs to characterize the implication of these interactions in several processes that include chromatin remodeling, mRNA and protein stability, transcription, and mRNA turnover [26]. Accumulating evidence has demonstrated that microRNAs (miRNAs), which are non-coding RNA molecules of around 22 nucleotides, and long non-coding RNAs, which are longer than 200 nucleotides, interact to regulate their own expression and the expression of mRNAs through several molecular mechanisms [27,28,29,30]. MicroRNAs can silence cytoplasmic mRNAs by triggering an endonuclease cleavage, by promoting translation repression or by accelerating mRNA deadenylation and decapping [31,32]. Thus, miRNA blocking by lncRNAs can directly inhibit these processes, promoting mRNA stabilization and inducing gene expression.

In this section, we will provide an overview of lncRNAs that prevent interaction of miRNAs with target mRNAs to protect them from miRNA-driven degradation. These lncRNAs are referred to as competitive endogenous RNAs (ceRNAs), decoys or sponges [33]. LncRNAs can act as ceRNAs by two different mechanisms. On the one hand, they are able to sequester miRNAs, avoiding their binding to target mRNAs, and on the other hand, they can directly interact with target mRNA transcripts to block miRNA binding sites in mRNA molecules. In this case, lncRNAs and miRNAs share common binding sites in the target mRNA. The interactions between miRNAs and ceRNAs are crucial for the regulation of several basal biological processes but have also been described to participate in different pathogenic conditions.

Most lncRNAs that block miRNA activity to enhance mRNA stability are transcribed from the opposite DNA strand to their paired (sometimes complementary) sense protein coding genes and are known as natural antisense transcripts (NATs). Even though there are some examples of NATs that code for proteins, such as Wrap53 [34] and DHPS [35], NATs usually lack protein coding potential [36] and are generally classified as lncRNAs [37]. NATs can alter their paired sense gene expression by exerting their effect at different levels, including transcription, mRNA processing, splicing, stability and translation [36,38,39,40,41]. Regarding the mechanisms by which NATs alter mRNA stability, the “Recycling hypothesis” suggests that reversible RNA duplex formation might trigger conformational changes in mRNA molecules, hindering the accessibility to RNA binding proteins (RBPs), both stabilizing and destabilizing RBPs, and miRNAs [37].

The best characterized NAT is probably the lncRNA *BACE1 antisense RNA* (*BACE1-AS*). This lncRNA is partially antisense to *BACE1,* a gene encoding the β-site amyloid precursor protein cleaving enzyme 1, which plays a crucial role in the pathophysiology of Alzheimer’s disease [42]. Interestingly, it has been shown that lncRNA *BACE1-AS* is markedly up-regulated in brain samples from patients with Alzheimer’s disease and promotes the stability of the *BACE1* transcript [43]. The lncRNA *BACE1-AS* regulates the expression of its sense partner through a synergistic mechanism that includes prevention of miRNA-induced mRNA decay and translational repression. Specifically, *miR-485-5p* and *BACE1-AS* share a common binding site in the sixth exon of *BACE1* mRNA transcript, and thus binding of *BACE1-AS* to this site avoids the interaction of *miR-485-5p*, hindering the translational repression and destabilization of *BACE1* mRNA by *miR-485-5p*, and eventually elevating *BACE1* levels [43].

Similar to *BACE1-AS*, *PTB antisense RNA* (*PTB-AS*) also modulates its sense mRNA stability by masking miRNA binding sites [44]. *PTB-AS* binds to the 3′ untranslated region (UTR) of *PTBP1*, a RBP that promotes gliomagenesis [45], and prevents *miR-9* binding, a neural-specific miRNA known to target the 3′ UTR of *PTBP1* for degradation [46].

In addition, a lncRNA named *FGFR3 antisense transcript 1* (*FGFR3-AS1*) which is antisense to *FGFR3* gene, was shown to be upregulated in an expression analysis performed in tumorigenic tissue from patients with osteosarcoma, when compared to non-cancerous tissue [30]. Bioinformatic analysis indicated that *FGFR3-AS1* and *FGFR3* formed a “tail-to-tail” fully complementarity pairing pattern composed of 1053 nucleotides, suggesting a potential regulatory effect of *FGFR3-AS1* in the expression of the *FGFR3* gene. In silico results were confirmed by RNA protection assays that showed that the non-overlapping part of *FGFR3* mRNA was totally digested, but the overlapping 3′UTR of *FGFR3* mRNA was protected from RNase digestion. Moreover, the authors showed that this antisense pairing between *FGFR3-AS1* and *FGFR3* mRNA upregulated *FGFR3* expression by increasing *FGFR3* mRNA stability. Interestingly, many miRNAs have been reported to bind to the 3′UTR of *FGFR3*, inducing *FGFR3* mRNA degradation [47]. Thus, the antisense pairing between *FGFR3-AS1* and the 3′UTR of *FGFR3* might block potential miRNA binding sites, protecting *FGFR3* from miRNA-induced degradation and/or translation inhibition. However, whether *FGFR3-AS1*-driven *FGFR3* mRNA stabilization occurs through this mechanism remains to be clarified.

*Paxillin antisense RNA 1* (*PXN-AS1*), a lncRNA overlapping *PXN* mRNA, was identified after discovering alternative splicing events on a transcriptome sequencing analysis of a hepatocellular carcinoma (HCC) cell line with stable deletion of *Muscleblind-like-3* (*MBNL3*), an oncofetal splicing factor. Two main transcripts were identified: lncRNA *PXN-AS1-L* (containing exon 4) and lncRNA *PXN-AS1-S* (lacking exon 4). Both, *PXN-AS1-L* and *PXN-AS1-S*, were preferentially expressed in the cytoplasm, but had different regulatory effects on the expression of the *PXN* transcript. While *PXN-AS1-L* upregulated PXN protein, *PXN-AS1-S* downregulated it. Interestingly, *PXN-AS1-L* upregulated *PXN* mRNA by preventing *miRNA-24*-AGO2 complex binding to the 3′UTR of *PXN* mRNA [48].

The lncRNA *Sirt1 antisense* (*Sirt1-AS*) is transcribed from the *Sirt1* antisense strand and has been shown to interact with *Sirt1* mRNA, forming an RNA duplex that increases stability of its paired transcript, prolonging its half-life up to 10 h and eventually augmenting SIRT1 protein expression [49]. Using luciferase assay experiments it was shown that *Sirt1-AS* lncRNA interacted with the 3′UTR of the *Sirt1* mRNA transcript. This interaction masked *miR-3a* binding sites, avoiding *miR-3a*-driven *Sirt1* mRNA degradation. Interestingly, SIRT1 is a NAD-dependent class III protein deacetylase, which regulates the balance between myoblast proliferation and differentiation, and plays a crucial role in muscle formation [50]. Thus, lncRNA *Sirt1-AS* might participate in myogenesis by blocking *miR-34a* binding to *Sirt1* mRNA which turns in increased SIRT1 protein and increased myoblast proliferation.

Similar to *Sirt1-AS*, a lncRNA named *Urothelial Cancer Associated 1* (*UCA1*) has been shown to regulate mRNA stabilization through directly binding to 3′UTRs of target mRNAs to protect them from miRNA-mediated degradation [51].

Other mechanisms by which lncRNAs block the effect of miRNAs on mRNA degradation are the ones described in the “Competing endogenous RNA” hypothesis, in which lncRNAs compete with miRNAs or RBPs to bind the same common target sequence in mRNAs. Some examples of lncRNAs that act as ceRNAs are described in the following paragraphs.

In addition to the ability to prevent miRNA-induced degradation by binding to mRNA transcripts, it has been described that *UCA1* can also control mRNA stabilization and gene expression by sponging miRNAs that negatively regulate gene expression [51]. In this specific case, lncRNA *UCA1* was implicated in the progression of colorectal cancer through its capacity to control a ceRNA network that fostered upregulation of several genes, including *ANLN*, *BIRC5*, *IPO7*, *KIF2A*, and *KIF23*.

*OIP5 Antisense RNA 1* (*OIP5-AS1*) is the mammalian homolog of *Cyrano* gene in zebrafish and it is important for controlling neurogenesis during development [52]. It is located upstream of the *OIP5* sense gene, but they do not overlap. It is known to act as a ceRNA for *miR-143-3p* in cervical cancer (CC) cells, sustaining the expression of *miR-143-3p*-targets, *ITGA6* [53] and *SMAD3* [54], and promoting proliferation, migration and invasion of CC cells [53,54].

Another lncRNA acting as a ceRNA that affects mRNA stability is *MACC1 Antisense RNA 1* (*MACC1-AS1*), an intronic antisense lncRNA located between the fourth and fifth exon of *MACC1*, a transcriptional regulator of epithelial-mesenchymal transition (EMT) [55] that enhances gastric tumor progression [56]. It shares binding sites for *miR-384* and *miR-145-3p* within *PTN* and *c-Myc* transcripts respectively, which are two well-known oncogenic genes [57]. Similar to lncRNA *OIP5-AS1*, *MACC1-AS1* has the capacity to sequester *miR-384* and *miR-145-3p*, sustaining the stability of *PTN* and *c-Myc* mRNAs, and promoting cell proliferation and tumorigenesis.

Other lncRNA that also acts as a miRNA sponge is lncRNA *PTENP1pg1* [58]. *PTENpg1* controls the expression of the tumor suppressor gene *PTEN*, and thus, plays a crucial role in tumorigenesis processes. Interestingly, antisense to this *PTENP1pg1*, there is another lncRNA named *PTENP1pg1-AS*, which has two isoforms, alpha and beta. While the alpha isoform functions in *trans* and epigenetically modulates *PTEN* transcription by the recruitment of DNMT3a and EZH2, the beta isoform interacts with *PTENpg1* through an RNA:RNA pairing interaction, affecting PTEN protein output via changes of *PTENpg1* stability and microRNA sponge activity.

It is also worth mentioning the lncRNA *uc.173* that inhibits miRNA function through a molecular mechanism that implies posttranscriptional reduction of a pri-miRNA. This lncRNA is transcribed from an ultraconserved region (UCR) in human chromosome 3. UCRs represent conserved sequences of the human genome that are likely to be functional but do not have coding potential [59]. RNA molecules transcribed from UCRs originate from genomic regions located in both intra- and intergenic regions with almost perfect evolutionary conservation in most of the mammalian genomes, suggesting that may have a key function in cell physiology and pathogenic processes [59]. Indeed, lncRNA *uc.173* has been described to be implicated in intestinal mucosal cell growth and renewal [60]. This lncRNA, which is highly expressed in intestinal mucosa, stimulates intestinal epithelial cell renewal by downregulating *miRNA195* expression through posttranscriptional reduction of *pri-miR-195*. Although the precise molecular mechanisms by which this lncRNA destabilizes the *pri-miR-195* transcript are unknown, it seems that the process is achieved through a direct lncRNA-mRNA interaction that enhances the degradation of *pri-miR-195* transcript. Downregulation of *miRNA195* by lncRNA *uc.173* results in upregulation of genes implicated in intestinal epithelium growth [60].

Finally, another interesting example is the tumor-promoting lncRNA *ncNRFR* (*non-coding Nras functional RNA*). This lncRNA contains a 22-nucleotide sequence that is identical to miRNA *let-7a* and differs from other miRNAs (*let-7b*, *let-7c*, *let-7d*, *let-7e*, *let-7f*, *let-7g*, *let-7i*, and *miR-98*) in only 1–4 nucleotides [61]. Overexpression of *ncNRFR* in a cell line of colon epithelial cells increased the activity of a heterologous reporter bearing a miRNA *let-7* target site, suggesting that *ncNRFR* lowered miRNA *let-7* function. The miRNA *let-7* is a tumor suppressor that inhibits the expression of several oncogenes, and thus tumorigenic function of *ncNRFR* might be linked to its ability to suppress the action of miRNA *let-7* upon endogenous target mRNAs. The molecular mechanisms by which *ncNRFR* blocks *let-7* remain to be clarified, although taking into account the high homology in the sequence of these two ncRNAs, it is plausible to think that *ncNRFR* might directly compete with *let-7* to bind target mRNA transcripts and inhibit *let-7*-mediated mRNA degradation.

In summary, during the last few years it has become apparent that there is a significant crosstalk between miRNAs and lncRNAs in the regulation of gene expression. Indeed, various ceRNAs have already been identified and their capacity either to sequester miRNAs or to block miRNA binding to target mRNAs has been widely described. Sequestration or blocking of miRNAs by lncRNAs implies a reduced interaction of the miRNAs with their target mRNAs, which eventually turns into increased mRNA stability and expression. Although this field of research has just started to emerge, future studies analyzing the interaction between these two non-coding molecules will explain many of the “unknowns” that still linger regarding the regulation of gene expression, both in basal and pathogenic conditions.

## 3. Interaction between lncRNAs and RBPs in mRNA Stabilization

It is well established that AU-rich elements (AREs) [62] and GU-rich elements (GREs) [63] are distinct sequence elements in the 3′-UTR of mRNAs. These regions are among the most common determinants of RNA stability in mammalian cells by which various RNA binding proteins (RBPs), including both stabilizing and destabilizing factors bind to, thereby modulating mRNA stability and/or translational efficiency [64]. There exist hundreds of different RBPs with a diverse number of functions through distinct RNA binding domains to which proteins bind and affect RNA fate [65].

A wide variety of research works have shown how RBPs directly bind to mRNA to accelerate mRNA decay or affect translation (increasing or blocking the processes) [64,66]. Interestingly, more and more lncRNAs are being described to also bind RBPs [67,68,69]. One of the best studied lncRNAs, *Xist*, can form ribonucleoprotein complexes (RNPs) in the nucleus to affect target gene transcription regulation [70]. It has been shown that lncRNAs can also be cytoplasmic and bind RBPs to affect other mRNA metabolism processes such as mRNA stability and turnover [71,72,73]. Depending on which factors interact with a given lncRNA, this could increase or decrease the targeted mRNA.

In some cases, lncRNAs bind to mRNA transcripts and help to recruit RBPs (stabilizing or destabilizing) affecting mRNA levels. For example, the *LncRNA-assisted stabilization of transcripts* (*LAST*) can stabilize *CCND1* mRNA through protection against nuclease activity by promoting the interaction between the RBP named CNBP and the 5′UTR of *CCND1* mRNA [74]. In other cases, lncRNAs prevent RBP and target mRNAs interaction by binding the mRNA transcript. This is the case of the lncRNA *Sros1*, which blocks the binding of *Stat1* mRNA to the RBP CAPRIN1, stabilizing the *Stat1* mRNA [75], and of lncRNA *7SL*, which interacts with the 3′UTR of *TP53* mRNA, thereby preventing HuR binding and repressing *TP53* translation [76].

Another example is *PDCD4 Antisense RNA 1* (*PDCD4-AS1*)*,* a NAT affecting stability of *PDCD4*, which is a tumor suppressor coding gene implicated in breast cancer (BC) [77]. In a study by Jadaliha et al. [78], overlapping regions between *PDCD4-AS1* and *PDCD4* were reported. Thus, *PDCD4-AS1* and *PDCD4* mRNA were found to form an RNA duplex, inducing an increase in *PDCD4* mRNA stability. In this case, RNA duplex formation prevented the interaction between *PDCD4* mRNA and HuR [79]. Although HuR usually acts as a stabilizing protein, it has been shown that HuR can form a complex with KSRP to destabilize mRNA molecules and induce a significant reduction in specific protein levels [79].

It is also possible that lncRNA-RBPs interactions influence downstream target mRNA. In turn, *LINC00324* [80], *TRPM2 Antisense RNA* (*TRPM2-AS*) [81], lncRNA *MY* [82], lncRNA *MEG3* [83], lncRNA *Gadd7* [84], lncRNA *FIRRE* [85], or lncRNA *H19* [86] among others, can bind different RBPs (both stabilizing and destabilizing), and thus affect target mRNA decay.

There are also lncRNA-RBP interactions that would indirectly affect mRNA. There are some lncRNAs that modulate RBP activity and hence, will affect downstream mRNA levels. LncRNA *NORAD* has been described to sequester PUMILIO proteins, which are key regulators for many mRNA stability and translation processes. Thus, *NORAD*-PUMILIO interaction represses mRNA stability and translation of target mRNAs [87]. LncRNAs *OCC1* and *OIP5-AS1* have also been described to bind HuR (a RBP that binds to thousands of mRNAs). While *OCC1* enhances HuR degradation [88], *OIP5-AS1* functions as a sponge for HuR and prevents binding to its targets [89]. In the case of *treRNA1*, this lncRNA downregulates the expression of E-cadherin by suppressing the translation of its mRNA. *TreRNA1* forms an RNP complex that, in turn, binds to eIF4G1 (an initiation factor of protein synthesis) affecting translation of the target mRNA [90].

It is also known that RBPs can influence lncRNA stability, that could also affect lncRNA function and target mRNAs at different levels. *LincRNA-p21* interacts with target *CTNNB1* and *JUNB* mRNAs and inhibits their translation efficiency. However, HuR RBP can inhibit the expression of *lincRNA-p21* by inducing its degradation, which promotes the binding of HuR to *CTNNB1* and *JUNB* mRNAs enhancing their translation, thus increasing the levels of these proteins [91]. The cytoplasmic RBP HuD can also increase the stability of *BACE1-AS* to further influence target mRNA stability [92].

However, lncRNAs can also reduce mRNA stability, making transcripts prone to degradation. One mechanism by which this happens is the Staufen 1 mediated decay. Staufen 1 (STAU1) protein binds the 3′UTR that contain duplex RNA structures to mediate mRNA decay and regulate gene expression [93]. LncRNAs have been found to form STAU1 binding sites by interacting with the 3′UTR of coding genes, thus downregulating their expression [94]. *TINCR* lncRNA was first found to bind STAU1 protein in the context of epidermal differentiation [95]. Further studies performed in gastric cancer, confirmed the binding of *TINCR* to STAU1 protein and found that this interaction induced the STAU1 mediated decay of *KLF2* mRNA. *KLF2*, which induces apoptosis, was described to be reduced in the cancer tissues, opposite to what happens to *TINCR*. Thus, interaction of *TINCR* lncRNA with STAU1 in cancer cells induces the degradation of *KLF2*, preventing apoptosis and contributing to the oncogenic potential of gastric carcinoma [96]. Additionally, other mechanisms of lncRNA-mediated mRNA degradation have also been described. As it is the case of *aHIF* antisense lncRNA, which overlaps the 3′UTR of *HIF1a* coding gene, and has the ability to destabilize *HIF1a* mRNA, subsequently decreasing HIF-1α protein expression in response to chronic hypoxia [97]. Rossignol F. et al. hypothesized that this destabilization occurs via *aHIF*-mediated exposure of AU rich elements present in the 3′ UTR of *HIF1a* mRNA, although the molecular mechanisms by which the mRNA is degraded have not been described yet [98].

All these RNA-protein complexes rely mostly in RNA secondary and tertiary structures that allow the direct interaction between molecules. Therefore, impairment of RNA structure will affect binding and function of the complex, leading to dysregulation of the related pathways [65,99]. There is growing evidence about disease-associated SNPs affecting lncRNA structure [100,101]. Taking into account that many complex disease-associated SNPs are enriched within lncRNAs [102], identifying those lncRNAs and how their binding to RBPs is affected could help find key targets in the associated diseases. One example is *lnc13*, which regulates the stability of *STAT1* mRNA in pancreatic beta cells [103]. *Lnc13* was first discovered in the context of celiac disease, a chronic inflammatory disorder of the intestine, where it has a stability-unrelated function and it regulates gene expression in the chromatin [104]. However, Gonzalez-Moro I. et al. recently related *lnc13* with other autoimmune disorder, type 1 diabetes (T1D), as they found that upregulation of *lnc13* in pancreatic beta cells induces the activation of the pro-inflammatory STAT1 pathway promoting the production of downstream inflammatory chemokines. *Lnc13* was found to enhance STAT1 protein levels by stabilizing its mRNA via interaction with the protein PCBP2 (Poly(rC)-binding protein 2) in the cytoplasm. Viral infections, which have been proposed as triggering factors for T1D [105], were found to induce *lnc13* translocation from the nucleus to the cytoplasm, enabling the interaction of *STAT1* mRNA with PCBP2, which promotes the signaling events that will ultimately lead to pancreatic beta cell destruction and T1D development [103].

Finally, the cell specific expression and functions of lncRNAs should be taken into consideration as this broadens the pathways that can be affected by lncRNA function. For example, *Linc-RoR* interacts with both hnRNP I (stabilizing factor) and AUF1 (destabilizing factor), with an opposite consequence in their interaction with *c-Myc* mRNA [106]. Alternatively, lncRNA *Epr* changes *Cdkn1a* gene expression by affecting both its transcription and mRNA decay through its association with the transcription factor SMAD3 or the RBP KHSRP, respectively. KHSRP is predominantly an mRNA decay promoting factor in epithelial cells and the interaction with *Epr* blocks its ability to induce decay of *Cdkn1a* mRNA [107,108].

All these mechanisms show the importance of studying lncRNA regulatory roles in mRNA stability and turnover, but also demonstrate the intricate work beyond studies for lncRNA functional characterization.

## 4. LncRNAs, Epitranscriptomic Changes and mRNA Stability

RNA modifications have been recently involved in the regulation of mRNA stability and it has been stated that the regulation of mRNA stability through RNA modification is a crucial step for the tight regulation of gene expression [109]. N6-methyladenosine (m^6^A) methylation is the most prevalent RNA modification in mRNAs and noncoding RNAs, and it has been involved in a wide range of RNA metabolic processes, including stability [110].

YTHDF2, an m^6^A reader protein, has been described to selectively bind to m^6^A-containing mRNAs, resulting in the localization of bound mRNAs from the translatable pool to cellular mRNA decay sites, such as processing bodies [110]. In contrast to the mRNA-decay-promoting function of YTHDF2, insulin-like growth factor 2 mRNA-binding proteins (IGF2BPs) promote the stability and storage of target mRNAs in an m^6^A-dependent manner [111]. The opposite role of IGF2BPs versus YTHDF2 imposes an additional layer of complexity on m^6^A function. IGF2BPs and YTHDF2 may recognize different targets or compete for the same m^6^A sites to fine-tune expression of shared targets through controlling the balance between mRNA stabilization and decay [111]. On the other hand, another m^6^A reader protein, YTHDF1, actively promotes protein synthesis by interacting with the translation machinery [112]. Altogether, YTHDF2 and IGF2BPs control the lifetime of target transcripts, whereas YTHDF1-mediated translation promotion increases translation efficiency.

In this context, there are few works describing lncRNAs influencing m^6^A-mediated mRNA stability. *GAS5-AS1* interacts with the tumor suppressor *GAS5* and increases its stability by influencing the interaction between *GAS5* mRNA and the RNA demethylase ALKBH5 leading to a decreased *GAS5* m^6^A methylation. Moreover, it was shown that m^6^A-mediated *GAS5* mRNA degradation relies on YTHDF2-dependent pathway [113]. *LINC00470* associates with *PTEN* mRNA and suppresses its stability through interaction with the m^6^A writer METTL3. In addition, *LINC00470*-METTL3-mediated *PTEN* mRNA degradation also relies on YTHDF2 [114]. Lastly, *LIN28B-AS1* is able to regulate mRNA stability of *LIN28B* by directly interacting with IGF2BP1 but not with *LIN28B*, as IGF2BP1 affects *LIN28B* mRNA stabilization [115].

In the context of mRNA-lncRNA interactions, lncRNA *LNC942*, upregulated in breast cancer, has been described to interact with the methylase METTL4 driving it to the mRNA of target genes *CXCR4* and *CYP1B1*. These two genes are involved in breast cancer initiation and progression, and their methylation augments the stability of the mRNA molecules, which results on higher protein levels and induction of tumorigenesis [115] probably due to an increased interaction with IGF2BP and YTHDF1 readers.

## 5. Concluding Remarks and Future Prospects

Correct tuning of mRNA stability is a crucial process to maintain appropriate homeostasis, and thus its dysregulation may lead to the development of several pathologies, including cancer. Stability of mRNA molecules is tightly regulated by several mechanisms, including the action of lncRNA molecules. During the last few years, lncRNAs have been implicated in the modulation of mRNA stability and several mechanisms of action have been described. On the one hand, they can prevent miRNA- and RBP-binding to target mRNAs by blocking target binding sites through direct lncRNA-mRNA interaction. On the other hand, they can sequester miRNAs and RBPs to avoid their interaction with target mRNAs, or to inhibit RBP-driven posttranscriptional modifications that affect mRNA stability. Thus, lncRNAs have emerged as crucial regulators of mRNA stability, another molecular mechanism by which these non-coding molecules participate in the regulation of gene expression.

Taking into account that lncRNAs play important roles in the regulation of mRNA stability, the functional characterization of the molecular mechanisms by which these non-coding molecules participate on mRNA equilibrium maintenance will open the door to the development of new lncRNA-based strategies to modify mRNA half-life and subsequent protein expression. Additionally, the functional understanding of lncRNAs that regulate mRNA stability in non-mammalian organisms as *Drosophila melanogaster* or zebrafish, which are easier to genetically manipulate, will help find human orthologous lncRNAs important in mRNA biology. As described formerly in this review, lncRNA-driven mRNA stability changes might impact several biological processes which are important, in both, health and disease. Thus, a better understanding of how lncRNAs act on mRNA stability will provide useful information for the development of new therapeutic strategies to treat and/or cure several diseases in which a dysregulated gene expression pattern is responsible of their development.

## Figures and Tables

**Figure 1 ncrna-07-00003-f001:**
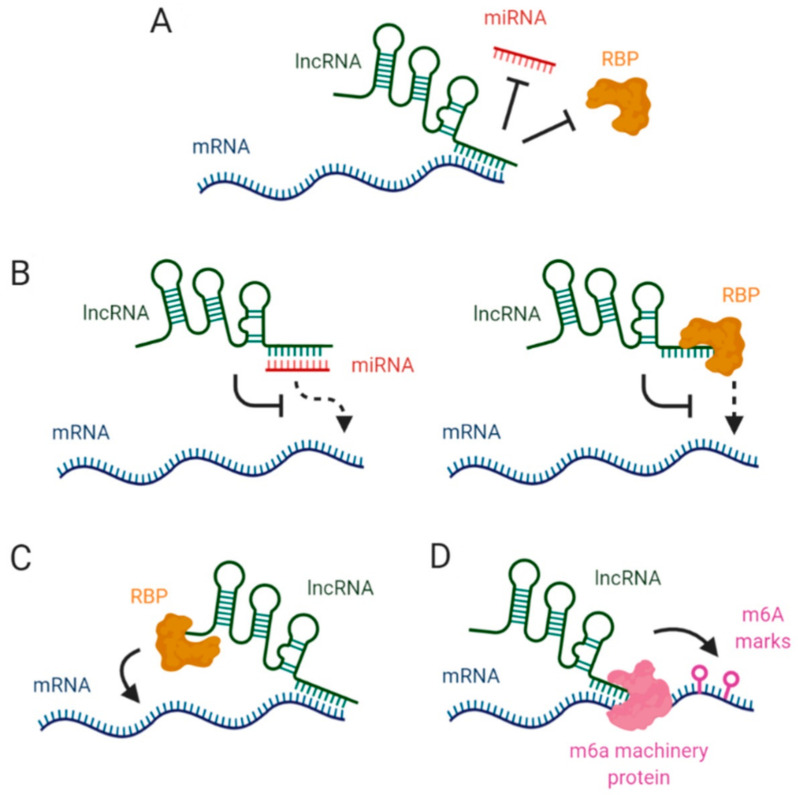
Mechanisms of action of lncRNA-mediated mRNA stability regulation. LncRNAs can modulate mRNA stability through different mechanisms: (**A**) Direct interaction with miRNA or RBP binding sites in target mRNA; (**B**) Sequestration of miRNAs or RBPs to avoid their interaction with mRNA molecules; (**C**) Acting as scaffolds to enhance RBP-mRNA interactions; (**D**) Interaction with m6A machinery to modulate m6A levels of target mRNAs.

## Data Availability

Not applicable.

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
