# Peer review of "The Role of lncRNAs in Gene Expression Regulation through mRNA Stabilization"

_ncrna, 2021, doi:10.3390/ncrna7010003_

Round 1

Reviewer 1 Report

Overall, I very much enjoyed reading this review.  I found it both informative and comprehensive and rather well organized.  My only suggestion is that the authors make a number of minor grammatical improvements:

Line 35: change “adequate against” to “adjust to”

Line 51: change “the last” to “the last few” or to “recent”

Line 72: same as for line 51

Line 73: change “their implications in” to “the implication of these interactions in”

Line 75: change “have” to “has”

Line 90: delete “, as we will unravel in the following lines”

LIne 106: change “synergic” to “synergistic”

Line 116: change “Otherwise” to “In addition”

Line 121: change “expression of” to “expression of the”

Line 122: change “that” to “that the” and “but” to “but the”

Line 135: change “of PXN” to “of the PXN”

Line 137: change “to 3’UTR” to “to the 3’UTR”

Line 142: change “of Sirt1” to “of the Sirt1”

Line 161: change “of OIP5” to “of the OIP5”

Line 192: change “into” to “in”

Line 193: replace the entire line with “Finally,”

Line 201: change “block” to “blocks”

Line 205: change “last” to “last few”

Line 216: here you define 3’UTR for the first time although you’ve mentioned these multiple times previously.

Line 219: delete “conduct”

Line 237: change “Other” to “Another”

Line 257: change “what” to “that”

Line 263: change “Nevertheless” to “However”

Line 265: It would be more accurate to say that STAU1 binds to 3’UTR regions that contain duplex RNA structures.

Line 316: change “translatable” to “the translatable”

Line 324: change “translation” to “the translation”

Line 344: change “the last” to “the last few” or “recent”

Author Response

Reviewer 1

Overall, I very much enjoyed reading this review.  I found it both informative and comprehensive and rather well organized.  My only suggestion is that the authors make a number of minor grammatical improvements:

Line 35: change “adequate against” to “adjust to”

Line 51: change “the last” to “the last few” or to “recent”

Line 72: same as for line 51

Line 73: change “their implications in” to “the implication of these interactions in”

Line 75: change “have” to “has”

Line 90: delete “, as we will unravel in the following lines”

LIne 106: change “synergic” to “synergistic”

Line 116: change “Otherwise” to “In addition”

Line 121: change “expression of” to “expression of the”

Line 122: change “that” to “that the” and “but” to “but the”

Line 135: change “of PXN” to “of the PXN”

Line 137: change “to 3’UTR” to “to the 3’UTR”

Line 142: change “of Sirt1” to “of the Sirt1”

Line 161: change “of OIP5” to “of the OIP5”

Line 192: change “into” to “in”

Line 193: replace the entire line with “Finally,”

Line 201: change “block” to “blocks”

Line 205: change “last” to “last few”

Line 216: here you define 3’UTR for the first time although you’ve mentioned these multiple times previously.

Line 219: delete “conduct”

Line 237: change “Other” to “Another”

Line 257: change “what” to “that”

Line 263: change “Nevertheless” to “However”

Line 265: It would be more accurate to say that STAU1 binds to 3’UTR regions that contain duplex RNA structures.

Line 316: change “translatable” to “the translatable”

Line 324: change “translation” to “the translation”

Line 344: change “the last” to “the last few” or “recent”

We are grateful to the Reviewer for his/her positive comments and for finding our review of interest. We have performed all the minor grammatical changes proposed by the Reviewer.

Reviewer 2 Report

The review is well written, informative, and comprehensive. It is a good review of extensive literature. But I have minor concerns and authors are suggested to address those before it is considered for acceptance.

  1. All the orthologous lncRNA candidates are not identified yet in mammals. Authors are suggested to include couple of sentences in the conclusion explaining how this is opening a window for researcher to dive into it.
  2. lncRNAs are expressed cell and tissue specific manner and Chowdhury et. al. (PMID: 29147069, PMID: 30687302) explained in their publication. Authors were suggested to include cell, tissue specific expression of lncRNAs and even all the splice variants are not expressed equally in single type of cells.
  3. Authors were suggested to include 2-3 sentences in the introduction section explaining how lncRNAs are functionally diverse.
  4. This review has typographical errors.

Author Response

Reviewer 2

The review is well written, informative, and comprehensive. It is a good review of extensive literature. But I have minor concerns and authors are suggested to address those before it is considered for acceptance.

We are grateful to the Reviewer for his/her comments, and much appreciate his/her thorough revision and relevant suggestions for improvement. Please find below the answers to his/her comments and the changes made in the manuscript accordingly.

  1. All the orthologous lncRNA candidates are not identified yet in mammals. Authors are suggested to include couple of sentences in the conclusion explaining how this is opening a window for researcher to dive into it.

We agree with the Reviewer and have included a brief section about the possibility of studying non-mammalian orthologous lncRNAs in the context of human lncRNA research (p.9, line 368).

  1. ncRNAs are expressed cell and tissue specific manner and Chowdhury et. al. (PMID: 29147069, PMID: 30687302) explained in their publication. Authors were suggested to include cell, tissue specific expression of lncRNAs and even all the splice variants are not expressed equally in single type of cells.

Following the Reviewer’s suggestion, we have included a sentence mentioning that lncRNA expression is cell/tissue/species-specific (p. 2, line 64).

  1. Authors were suggested to include 2-3 sentences in the introduction section explaining how lncRNAs are functionally diverse.

Following the Reviewer’s suggestion we have included a short paragraph describing the diverse functions of lncRNAs in the Introduction section (p.2, line 59).

  1. This review has typographical errors.

We thank the Reviewer for pointing out these errors, we have double checked the manuscript and corrected all the typographical errors.

Reviewer 3 Report

This is a well-written review on recent studies of lncRNA’s roles in regulating mRNA turnover. The authors did a good job by concisely describing extensive examples that support the various ways of lncRNAs in modulating mRNA stability.

This reviewer has the following points to help strengthen the manuscript:

  1. P2, line 54: “However, the regulation of mRNA stability and decay cannot be simplified…”. The authors may consider rewriting this statement. For instance, “However, the regulation of mRNA stability depends largely on how a three step process is modulated by regulatory factors, and thus these factor should be ….” This is because the three steps process is indeed the major or default way of triggering and degrading mRNAs in eukaryotes.
  2. P2, line 62: “interact with RNA binding proteins or miRNAs to avoid …”. Target mRNAs should also be included here as one of the lncRNA-interacting substrates, i.e., “interact with target mRNAs, RNA binding proteins or miRNAs to avoid …”.
  3. P3, line 79: “…by promoting translation repression or by accelerating mRNA decapping [27].” The deadenylation step, which is the first step of being accelerated by RISC complex [Chen et al. 2009. NSMB, 16: 1160], should be included here with the reference, i.e., “…by promoting translation repression or by accelerating mRNA deadenylation and decapping [27] [Chen et al. 2009. NSMB, 16: 1160]”.
  4. P3, line 101: there are two consecutive words of “antisense”. Delete one.
  5. P4, lines 141-143: This statement needs to be further elaborated as it is unclear what exactly the point the authors are trying to convey.
  6. P5, line 215: References should be given here regarding ARE, e.g., “Chen and Shyu, 1995, TiBS, 20: 465.“ and regarding GRE, e.g., “Vlasova and Bohjanen, RNA Biol 2008, 5, 201. “.
  7. HuR typically helps to stabilize mRNA (also pointed out by the authors, see the statement in line 252). In this particular case described by the authors, HuR forms a complex with a destabilizing ARE-BP, KSRP, that together promotes destabilization. Please elaborate and modify the original sentence because it's misleading as stated.
  8. P7, line 307: Please delete “tedious” and use a different word, e.g., “intricate”.
  9. P8, lines 327-328: The authors may want to reconsider this example, as it is described in a way to imply that GAS5-AS1 and ALKBH5 interaction only affects GAS5 and not others. It raises the question as to how is it possible that interfering with a general demethylase ALKBH5 could lead to only a specific reduction in GAS5 m6A modification without affecting other m6A-mRNA?
  10. P8, lines 335-339: Please elaborate here on how increased m6A modification of those two mRNAs augments their stability, which is at odds with the general belief that m6A binding by YTHDF2 usually induces mRNA decay.
  11. The last point from this reviewer is that the authors may consider write briefly about future prospects and directions of the study of lncRNAs on mRNA stability, which would be a nice way to end the review.

Author Response

Reviewer 3

This is a well-written review on recent studies of lncRNA’s roles in regulating mRNA turnover. The authors did a good job by concisely describing extensive examples that support the various ways of lncRNAs in modulating mRNA stability.

We are grateful to the Reviewer for his/her comments, and much appreciate his/her suggestions for improvement. Please find below the answers to his/her comments and the changes made in the manuscript accordingly.

This reviewer has the following points to help strengthen the manuscript:

  1. P2, line 54: “However, the regulation of mRNA stability and decay cannot be simplified…”. The authors may consider rewriting this statement. For instance, “However, the regulation of mRNA stability depends largely on how a three step process is modulated by regulatory factors, and thus these factor should be ….” This is because the three steps process is indeed the major or default way of triggering and degrading mRNAs in eukaryotes.

We have rewritten this sentence following the Reviewer’s suggestion.

  1. P2, line 62: “interact with RNA binding proteins or miRNAs to avoid …”. Target mRNAs should also be included here as one of the lncRNA-interacting substrates, i.e., “interact with target mRNAs, RNA binding proteins or miRNAs to avoid …”.

We have added “target mRNAs” in this sentence as suggested by the Reviewer.

  1. P3, line 79: “…by promoting translation repression or by accelerating mRNA decapping [27].” The deadenylation step, which is the first step of being accelerated by RISC complex [Chen et al. 2009. NSMB, 16: 1160], should be included here with the reference, i.e., “…by promoting translation repression or by accelerating mRNA deadenylation and decapping [27] [Chen et al. 2009. NSMB, 16: 1160]”.

We have modified this sentence to include the deadenylation process and added the reference accordingly.

  1. P3, line 101: there are two consecutive words of “antisense”. Delete one.

We thank the Reviewer for pointing out this mistake. We have removed the duplicated word.

  1. P4, lines 141-143: This statement needs to be further elaborated as it is unclear what exactly the point the authors are trying to convey.

We agree with the Reviewer that this statement was not clearly explained, and thus we have now modified it as follows:

“Using luciferase assay experiments it was shown that Sirt1-AS lncRNA interacted with the 3’UTR of the Sirt1 mRNA transcript. This interaction masked miR-3a binding sites, avoiding miR-3a-driven Sirt1 mRNA degradation. Interestingly, SIRT1 is a NAD-dependent class III protein deacetylase, which regulates the balance between myoblast proliferation and differentiation, and plays a crucial role in muscle formation [45]. Thus, lncRNA Sirt1-AS might participate in myogenesis by blocking miR-34a binding to Sirt mRNA which turns in increased SIRT1 protein and increased myoblast proliferation”

  1. P5, line 215: References should be given here regarding ARE, e.g., “Chen and Shyu, 1995, TiBS, 20: 465.“ and regarding GRE, e.g., “Vlasova and Bohjanen, RNA Biol 2008, 5, 201.

We have included these two references as suggested by the Reviewer.

  1. HuR typically helps to stabilize mRNA (also pointed out by the authors, see the statement in line 252). In this particular case described by the authors, HuR forms a complex with a destabilizing ARE-BP, KSRP, that together promotes destabilization. Please elaborate and modify the original sentence because it's misleading as stated.

The Reviewer is right and thus, we have removed that HuR is a stabilizing protein in line 252, and further describe its role as a destabilizing protein in line 250.

  1. P7, line 307: Please delete “tedious” and use a different word, e.g., “intricate”.

Following Reviewer’s suggestion, we have replaced “tedious” for “intrincate”.

  1. P8, lines 327-328: The authors may want to reconsider this example, as it is described in a way to imply that GAS5-AS1 and ALKBH5 interaction only affects GAS5 and not others. It raises the question as to how is it possible that interfering with a general demethylase ALKBH5 could lead to only a specific reduction in GAS5 m6A modification without affecting other m6A-mRNA?

We understand the Reviewer’s concern, and we have modified this part for better understanding. In this example, GAS5-AS1 interacts with GAS5 influencing its interaction with ALKBH5 which leads to a decrease in the methylation levels of GAS5 mRNA. The lncRNA GAS5-AS1 directly interferes with the methylation of GAS5 without affecting the general function of ALKBH5.

  1. P8, lines 335-339: Please elaborate here on how increased m6A modification of those two mRNAs augments their stability, which is at odds with the general belief that m6A binding by YTHDF2 usually induces mRNA decay.

Although it is true that the general belief has been that methylation induces mRNA decay by YTHDF2 binding it has been also described that some m6A readers (as IGF2BPs) can recognize and stabilize methylated RNAs (S Hun Han, et al., EMM, 2020). Thus, although the mechanism by which increased m6A levels stabilized CXCR4 and CYP1B1 mRNA and induce their protein levels were not described in this paper, we believe that methylation induces their interaction with IGF2BPs and/or YTHDF1 readers leading to stabilization and translation. We have included a sentence to clarify this issue in the corresponding section.

  1. The last point from this reviewer is that the authors may consider write briefly about future prospects and directions of the study of lncRNAs on mRNA stability, which would be a nice way to end the review.

Following Reviewer’s suggestion, we have added a brief paragraph describing the future prospects of the study of lncRNAs on mRNA stability in the last section, which is now named “Concluding Remarks and Future Prospects”.